# Targeting the Gut Microbiome to Treat Metabolic Dysfunction-Associated Fatty Liver Disease: Ready for Prime Time?

**DOI:** 10.3390/cells11172718

**Published:** 2022-08-31

**Authors:** Nicolas Lanthier, Nathalie Delzenne

**Affiliations:** 1Service d’Hépato-Gastroentérologie, Cliniques Universitaires Saint-Luc, UCLouvain, 1200 Brussels, Belgium; 2Laboratory of Gastroenterology and Hepatology, Institut de Recherche Expérimentale et Clinique, UCLouvain, 1200 Brussels, Belgium; 3Metabolism and Nutrition Research Group, Louvain Drug Research Institute, UCLouvain, 1200 Brussels, Belgium

**Keywords:** MAFLD, NASH, microbiota, microbiome, gut, prebiotic, probiotic, postbiotic, steatosis

## Abstract

Numerous studies show a modification of the gut microbiota in patients with obesity or diabetes. Animal studies have also shown a causal role of gut microbiota in liver metabolic disorders including steatosis whereas the human situation is less clear. Patients with metabolic dysfunction associated fatty liver disease (MAFLD) also have a modification in their gut microbiota composition but the changes are not fully characterized. The absence of consensus on a precise signature is probably due to disease heterogeneity, possible concomitant medications and different selection or evaluation criteria. The most consistent changes were increased relative abundance of Proteobacteria, Enterobacteriaceae and *Escherichia* species and decreased abundance of *Coprococcus* and *Eubacterium*. Possible mechanisms linking the microbiota and MAFLD are increased intestinal permeability with translocation of microbial products into the portal circulation, but also changes in the bile acids and production of microbial metabolites such as ethanol, short chain fatty acids and amino acid derivatives able to modulate liver metabolism and inflammation. Several interventional studies exist that attempt to modulate liver disease by administering antibiotics, probiotics, prebiotics, synbiotics, postbiotics or fecal transplantation. In conclusion, there are both gaps and hopes concerning the interest of gut microbiome evaluation for diagnosis purposes of MAFLD and for new therapeutic developments that are often tested on small size cohorts.

## 1. The Gut Microbiota: General Concepts and Interaction with Obesity, Diabetes and Metabolic Syndrome

Microbes located in the human gut are now considered key players in the metabolism of their host. These include bacteria but also other micro-organisms such as viruses, Archaea, Eukarya and fungi [1]. All these microbes are capable of interacting with each other but also with human cells. The commensal intestinal microbiota is essential to our health. It maintains the integrity of the intestinal barrier, participates to the defense against the invasion of pathogens and acts as modulator of the immune system, and it ensures the production of key metabolites such as bile acids, or short-chain fatty acids (SCFA) issued from non-digestible carbohydrates fermentation, which play a role in the regulation of inflammation, cell proliferation and mucus secretion. Conversely, alterations in the microbiota have been demonstrated to be associated with several diseases and correspond then to the “dysbiosis”. Among the microbial groups, bacteria have been mostly studied [1]. In 2007, Gram-negative bacterial constituents such as lipopolysaccharides (LPS) were highlighted as key factors driving the chronic low-grade inflammation characteristic of obesity and insulin resistance [2,3]. This endotoxemia stimulating toll-like receptor 4 (TLR4) receptor and its cofactor cluster of differentiation 14 (CD14) have established the link between the gut of obese patients and their insulin resistance state which is a central feature of the metabolic syndrome. Changes in the microbiota in obese patients have been first demonstrated at the phylum level, the ratio of Firmicutes to Bacteroidetes being associated, in the prime studies, to the metabolic disorders in obesity and diabetes [4]. However, even these characteristic changes have not been replicated. Nevertheless, the causal role of the microbiota in diabetes-related insulin resistance has finally been proven by fecal microbiota transplantation experiments from slim to obese subjects with the reference gold standard technique evaluation of hyperinsulinemic euglycemic clamp [5]. This evaluation showed a transient improvement in peripheral insulin resistance in obese patients who received microbiota from thin patients [5]. This effect was mainly related to elevated levels of butyrate from the transplanted microbiota [5]. This opens up the possibility that factors other than LPS and intestinal permeability are responsible for the metabolic homeostasis, as well as the concept of the “microbiome” which includes the microbiota and its by-products. Indeed, the revisited definition of the microbiome encompasses both the microorganisms and their “theatre of activity” (microbial structural elements such as polysaccharides, proteins and nucleic acids, the surrounding environmental conditions and microbial metabolites/signal molecules) [6].

## 2. Metabolic Dysfunction-Associated Fatty Liver Disease: A Growing Welfare Disease

The prevalence of overweight and obesity has increased in the population and continues to rise as a slow epidemic [7]. The prevalence of obesity in children and adolescents is also increasing [7]. Alongside this increase, non-alcoholic fatty liver disease (NAFLD), now clearly also called “metabolic dysfunction-associated fatty liver disease” (MAFLD), has emerged [8,9,10]. The disease is defined by the presence of excess fat in the liver, i.e., more than 5% of steatotic hepatocytes on histological analysis or other indirect markers of hepatic fat overload. Different stages of the disease are possible: isolated steatosis or the development of a more severe form where steatosis is associated with lobular inflammation and hepatocyte damage called hepatocyte ballooning. The coexistence of steatosis, lobular inflammation and hepatocyte ballooning defines non-alcoholic steatohepatitis (NASH). This disease can progress to progressive liver fibrosis and lead to cirrhosis. The disease is of course common in obesity, metabolic syndrome or type 2 diabetes [11,12,13]. Other risk factors are also described [14,15,16]. Among these, the gut-liver axis is involved not only in the initiation but also in the progression of MAFLD [17,18].

## 3. The Liver: The First Victim of Changes from the Digestive Tract but also a Culprit? The Concept of a Bidirectional Gut-Liver Axis

The liver is the first organ to be exposed to microbes (microbiota) from the gastrointestinal tract, but also to its components and metabolites (microbiome). Indeed, the liver receives portal venous blood—representing about 70% of the blood supply— from the gastrointestinal tract. However, the relationship between the gut and the liver is bidirectional. As an illustration, the liver produces bile which is secreted via the bile duct into the small intestine. Animal experiments have shown that ligation of the common bile duct, for example, leads to profound changes in the intestinal microbiota and intestinal permeability [19]. In patients with cirrhosis (regardless of the cause), there is also an increase in the translocation of intestinal bacteria and an increase in circulating bacterial DNA fragments [20]. When compared to healthy individuals, the composition of the microbiota of patients with cirrhosis showed a significant decrease in *Bacteroidetes* and an increase in *Proteobacteria* and *Fusobacteria* [21]. This indicates that the results of association studies between gut microbiota and liver diseases should be interpreted with some caution: changes in the microbial composition of the gastrointestinal tract are not necessarily the cause of the changes observed in the liver. Liver disease itself also affects the composition of the gut microbiota.

## 4. Evidence for a Causal Role of the Gut Microbiota in the Pathogenesis of MAFLD in Experimental Animal Models

There is considerable evidence that the gut microbiome can induce obesity, insulin resistance and liver steatosis. Firstly, germ-free animals do not develop obesity even when fed a high-fat or high-sugar diet [22]. Transplantation of gut microbiota from conventional normal animals to these germ-free animals results in adiposity and insulin resistance within two weeks [23]. Secondly, transplanting the microbiome of animals that develop steatosis and inflammation leads to the same liver abnormalities in the recipients, with the important caveat that there are “responder” and “non-responder” animals [24]. On the contrary, fecal transplantation from control animals to animals fed on a high-fat diet decreases liver and fat content and liver inflammation. This transplantation is associated with higher butyrate levels [25]. Thirdly, strategies based on the microbiome in order to manipulate it show interesting results in animals. These treatments include the administration of prebiotics, probiotics and synbiotics. The various data with this type of treatment are presented in a recent review manuscript [26]. With prebiotic, probiotic or synbiotic interventions, hepatic steatosis is often decreased with changes observed in hepatic gene expression (decreased lipogenesis and/or increased fatty acid oxidation) [26]. All these promising results obtained in animals therefore open the door to experiments in humans.

## 5. What Changes in the Gut Microbiota in Humans with MAFLD?

A large number of studies describe the composition of bacteria in fecal samples from patients with MAFLD. However, the results are very variable and sometimes contradictory. In general, a decrease in the diversity of the microbiota is observed in patients with MAFLD compared to control patients (Table 1) [27,28,29,30]. However, this change in diversity is not observed in patients with steatosis and obesity regardless of the stage of severity of the disease (Table 1) [27,31,32,33]. The changes usually observed in patients with steatosis compared to control individuals are an increase in the abundance of Proteobacteria at the phylum level, Enterobactariaceae at the family level and *Escherichia* at the genera level [34]. A decrease in Rikenellaceae and Ruminococcaceae at the family level, *Anaerosporobacter*, *Coprococcus, Eubacterium, Faecalibacterium* and *Prevotella* at the genera level is also observed [34]. When compared with patients with NASH or advanced fibrosis, individuals with more severe liver disease display decreased Gram-positive bacteria, Firmicutes phylum, Prevotellaceae family and *Prevotella* genus and increased abundance of Gram-negative bacteria, increased Enterobacteriaceae family (*Bacteroides, Ruminococcus* and *Shigella* genera) [34]. The relative abundance of fecal *Clostridium sensu stricto* is also significantly decreased in MAFLD patients with increased liver elasticity compared to MAFLD patients with less severe liver disease [31] and in NASH patients compared to controls [27]. We can therefore imagine a microbial signature that would reflect the stage of the liver disease. All these changes must be interpreted with some caution as studies vary according to diagnostic criteria, the presence of co-factors (obesity, diabetes, concomitant medications, statistics used). Interestingly, these changes are not fixed and a shift toward a healthy microbiome is observed in longitudinal studies on adults with NASH and clinical improvement over time [35].

Most research to date has focused on changes in bacterial composition [36]. However, recent data are available on fungi [37] and viruses [38] in patients with MAFLD also showing differences in gut mycobiome and virome depending on the stage of the disease.

## 6. What Are the Potential Mechanisms That Explain the Link between Gut Dysbiosis and MAFLD?

### 6.1. Microbiota Changes in MAFLD: Cause or Consequence?

The causal role of digestive tract disturbances leading to steatosis in humans is much less clear than in animals. The intestinal microbiome can lead to changes in the liver, but liver alterations (such as cirrhosis or cholestasis) also lead to changes in the gut microbiota and its permeability [43]. The question of cause and effect in the field of steatosis also exists between the liver and other metabolically active tissues such as the adipose tissue or the muscle. In a dysmetabolic context, many changes are observed in the intestines, the liver, the adipose tissue, but also in the muscles and brain in particular, and it is not clear where the trigger lies. The answer is probably not unique because, unlike in animal models, the human disease can be caused by variable and sometimes interconnected phenomena such as a diet rich in fat, a diet rich in sugar, a high level of sedentary activity, a variable involvement of alcoholic beverages, different genetic background, the impact of some drugs, the level of insulin resistance, etc. [14]. Despite this, there are general mechanisms based on interesting translational research that can explain the role of the intestinal microbiome in the pathogenesis of steatosis in humans. Those include changes in intestinal permeability, translocation of bacterial products, bile acid modulation and production of other bacterial metabolites.

### 6.2. Changes in Intestinal Permeability, Translocation of Bacterial Products and Activation of Innate Immunity and Oxidative Stress

Some patients with steatosis have increased intestinal permeability [44]. Microbial compounds such as lipopolysaccharides (LPS or endotoxin, specifically of Gram-negative bacteria), known as pathogen associated molecular patterns (PAMPs), can then pass through the intestinal wall and reach the liver via the portal vein. In the liver, these compounds can activate TLRs (TLR4), leading to hepatic inflammation, and in particular to the activation of liver resident macrophages (Kupffer cells), the first immune cells implicated in hepatic insulin resistance and in the development of steatohepatitis [17,45,46,47]. PAMPs are also bound by other pattern recognition receptors (PRRs) inducing mitochondrial reactive oxygen species (ROS) production and nuclear gene expression [48]. Chronic oxidative stress is one of the key mechanisms responsible for inflammation and disease progression [48]. Oxidative stress markers such as urinary 8-iso- prostaglandin F2α and serum soluble NOX2-derived peptide levels are indeed correlated with serum cytokeratin 18-M30 levels, a marker of liver damage [49]. However, an increase in intestinal permeability has not been demonstrated in all patients with steatosis [50]. In ALD, a decrease in immune defenses in the digestive tract has been shown to play a major role in the spread of bacterial fragments from the damaged intestinal mucosa [51,52,53]. This impaired gut immunity is apparently specific to ALD and not present in MAFLD patients [53]. 

### 6.3. Bile Acid Modulation

The liver produces an average of 500 milliliters of bile per day. Bile acids are produced by the liver and play an indispensable role in the emulsification of dietary fat. However, bile acids also act in the regulation of lipid metabolism and glucose homeostasis. Primary bile acids (cholic acid and chenodeoxycholic acid) are synthesized from cholesterol through the action of a cytochrome P450 enzyme, cholesterol 7α-hydroxylase (CYP7A1), conjugated in hepatocytes (with glycine and taurine) and reach the duodenum after being secreted in the bile ducts (Table 2). Many intestinal bacteria are capable of deconjugating (such as Clostridium) and dehydroxylating (such as Bacteroides) bile acids into deoxycholic and lithocholic acid (Table 2). When fibrosis increases, there is a decrease of the proportion of the primary bile acids, especially chenodeoxycholic acid [54]. The primary chenodeoxycholic acid is the most important activator of the farnesoid X receptor (FXR) while secondary bile acids are Takeda G-protein-coupled receptor 5 (TGR5) agonists. Those two receptors are now important targets for NASH treatment [55,56]. Since bile acids are metabolized by microbes in the gastrointestinal tract, it is therefore understandable that bacterial changes can modulate bile acid proportions and thus affect host metabolism.

### 6.4. Production of Bacterial Metabolites: Short-Chain Fatty Acids, Indole, Ethanol, Phenylacetic Acid and Trimethyl-5-aminovaleric Acid

Short-chain fatty acids (SCFA) are the main products of the fermentation of dietary fibers and carbohydrates by intestinal (colonic) bacteria (Table 2) [26]. These include butyrate, propionate and acetate. The effects of these short-chain fatty acids on the metabolism are varied and sometimes contradictory [26]. Butyrate and propionate reduce intestinal inflammation and maintain the integrity of the intestinal barrier. Butyrate also increases GLP-1 secretion, inhibits lipogenesis and increases fatty acid oxidation [26]. Butyrate also has effects on muscle, decreasing insulin resistance [64]. Propionate also inhibits lipogenesis while acetate is a substrate for lipogenesis [26]. Acetate is also a metabolic stimulus involved in the activation of T cells, triggering auto-aggression in NASH [65]. Interestingly, these SCFA are among the breath volatile metabolites (BVM) that can partly be measured in exhaled air [66]. We can therefore imagine respiratory tests to assess the severity of the liver disease or the impact of an intervention. Butyrate, for example, is significantly increased in subjects after taking dietary fiber (chitin glucan) [67]. Animal experiments have shown that *Roseburia* is able to promote butyrate production from chitin [68]. 

Another bacterial metabolite with a beneficial effect on the liver is indole, derived from dietary tryptophan (Table 2). Indole derivatives (such as indole-3-acetic acid or IAA) act as ligands for the aryl hydrocarbon receptor (AhR), present on epithelial and immune cells and thus playing a role on intestinal barrier function and immunity. Subjects with obesity or diabetes have less fecal IAA compared to non-obese subjects [69]. In animals, indole supplementation has a beneficial effect not only on gut permeability and immunity [60] but also on liver macrophage activation [58,59]. Importantly, liver macrophage activation is the first step associated with MAFLD development in humans [45] and associated with hepatic insulin resistance [17].

Other microbial metabolites could have the opposite effect and contribute to liver damage. Microbial fermentation of polysaccharides in the gut can lead to actaldehyde and finally to endogenous ethanol production (Table 2), even in the absence of alcohol consumption. NASH patients exhibit significantly elevated blood ethanol levels compared to healthy controls [40]. *Klebsiella pneumoniae* and Enterobacteriaceae including *Escherichia* are notably involved in this production (Table 2) [40,61]. This endogenous ethanol production by the gut microbiota can contribute to the development but also to the progression of MAFLD [40]. *Klebsiella pneumoniae* has been identified in more than half of the patients with steatosis in some series. Fecal transplantation of *Klebsiella pneumoniae* in animals leads to the development of steatosis [61].

Finally, gut microbes also produce phenylacetic acid (PAA), a bacterial metabolite of phenylalanine and trimethyl-5-aminovaleric acid (TMAVA) from trimethyllysine that induces steatosis (Table 2) [39,62]. Interestingly, its level is associated with steatosis in humans and the chronic treatment with these compounds in animals trigger steatosis [39,62]. Succinate, produced by the metabolism of non-digestible polysaccharides, plays a controversial role [63].

## 7. Interventional Studies Targeting the Microbiota in MAFLD

Possible interventions targeting the gut microbiota in humans include the administration of antibiotics, probiotics, prebiotics, synbiotics or fecal microbiota transplantation. The use of postbiotics, although little tested in humans, is also presented.

### 7.1. Antibiotics

The most studied antibiotic is rifaximin (Table 3). Although not all studies agree on a positive effect [70,71,72] (Table 3), a randomised placebo-controlled study with long-term intervention in patients with histologically proven NASH showed beneficial effects on both transaminase levels, disease activity assessed by circulating levels of cleaved cytokeratin 18 (CK-18) and insulin resistance [72].

### 7.2. Probiotics

Probiotics are live micro-organisms that provide a health benefit to the host when administered at adequate levels. Some data also show promising results (Table 3). The administration of *Bifidobacterium*, *Lactobacillus*, *Lactococcus*, *Propionibacterium* strains has been tested versus placebo in a 8-week intervention. A decrease in the fatty liver index, AST and γGT is noted, without impact on ALT level or liver elasticity [73]. A proof-of-concept prospective study shows the feasibility of culturing and administering *Akkermansia muciniphila* to humans [74]. Interestingly, the administration of its pasteurised form to overweight subjects induces indirect changes in liver status with a significant decrease in γGT and AST levels compared to controls, in addition to an improvement in insulin resistance [74].

### 7.3. Prebiotics

Prebiotics are substrates used by host micro-organisms that confer a health benefit. In a small placebo controlled trial on 7 MAFLD patients, the administration of oligofructose significantly decreased serum AST levels after 8 weeks (Table 3) [75]. The only study that included a biopsy after the procedure also included only a small number of patients [76]. In 8 patients, the histological severity score of non-alcoholic steatohepatitis decreased with oligofructose administration but only due to regression of the steatosis stage [76]. There was no impact on transaminases [76]. It is also important to note that the first comparison biopsy was sometimes performed 5 years before inclusion in the study [76]. Inulin, administered for 42 days, showed no beneficial effect versus placebo in a small study [77]. An increase in steatosis level at the end of the study compared to the beginning was also noted [77]. When given to patients with obesity and steatosis during 3 months, a significant decrease in AST and weight was noted compared to placebo [78]. Interestingly, the research group was able to identify patient-specific characteristics associated with a positive impact of inulin (Table 3) [79]. 

### 7.4. Synbiotics 

Synbiotics (combination of probiotics and prebiotics) have also been studied in patients with MAFLD. In a randomized, controlled, nonblinded trial, patients receiving *Lactobacillus*, *Bifidobacterium*, *Streptococcus thermophilus* strains and fructooligosaccharides for 12 weeks had a decrease in ALT and liver stiffness compared with placebo [80]. An indirect beneficial impact on liver tests and steatosis severity of ultrasound was noted in one study [81]. In another study, no indirect change in steatosis assessed by MRI or fibrosis assessed by elastography was found (Table 3) [82]. 

### 7.5. Postbiotics

The concept of postbiotics is based on the observation that the beneficial effects of the microbiota could be due to the secretion of various metabolites. Initially, postbiotics were defined as a substance released by or produced through the metabolic activity of the microorganism which exerts a beneficial effect on the host, directly or indirectly [85]. Because of the risks of infection or antibiotic resistance (probiotics strains that carry antibiotic resistance genes themselves [86]) associated with the administration of probiotics, there is some interest in postbiotics [85]. Many results are available in animals, with positive results, for example, on insulin sensitivity and adiposity in mice fed a high fat diet and treated with butyrate [64]. Among the postbiotics tested in humans, the most detailed results concern ursodeoxycholic acid (a secondary bile acid which can be considered as postbiotic [36]). Although its administration reduces transaminases and insulin resistance in some studies [87,88], it does not provide histological improvement [89]. The modified or semi-synthetic bile acids currently being tested [55] are not discussed here as they are not found in humans (norursodeoxycholic acid or obeticholic acid, for example). To date, other data on postbiotics in humans only concern butyrate. Butyrate supplementation does not increase blood butyrate levels, but it improves both peripheral and hepatic insulin sensitivity assessed by clamp studies in lean individuals without metabolic syndrome but not in individuals with metabolic syndrome [90]. This suggests a positive impact, but too little in the face of a pro-inflammatory trigger. Finally, we have to mention that, in 2021, the International Scientific Association of Probiotics and Prebiotics decided to change the definition of postbiotics and to apply the term to inanimate microorganisms and/or their components conferring a health benefit on the host [91]. The term would therefore no longer include purified metabolites alone, in the absence of cellular biomass [91]. The use (beneficial in a small group of subjects) of the pasteurised form of *Akkermansia muciniphila* [74] described above (Table 3) may therefore fit this new definition. This new definition has been critical because few studies using postbiotics actually use inanimate bacteria [92]. 

### 7.6. Fecal Microbiota Transplantation

Fecal microbiota transplantation has been tested in MAFLD patients. One infusion was not associated with decreased liver steatosis in one study [83]. In another, a 3-day infusion induced a modest, although significant decrease in steatosis severity (Table 3) [84].

## 8. Why Do We Observe Variable or Weak Responses in Interventional Studies?

Several factors are identified below that may be responsible for the inconsistent, weak or negative results obtained in the studies.

### 8.1. Varied Endpoints

The gold standard for the diagnosis of NASH and the estimation of its severity is a liver biopsy. Two histological endpoints for approval of new drugs have been designated: resolution of steatohepatitis without worsening of fibrosis and at least one-point improvement in fibrosis stage with no worsening of steatohepatitis [55,93]. However, biopsy remains a technique with certain limitations such as the analysis of only a small fragment of the entire liver, the cost, the invasiveness and the inter- and intra-individual variability in the analysis. The measurement of liver fat content via magnetic resonance imaging has been proposed and is now used in some early-phase phase NASH clinical trials [94,95]. In the field of pre, pro or synbiotic studies, the targets are not uniform. MRI or biopsy are not frequently used. 

### 8.2. Presence of Metabolic Cofactors or Concomitant Medications

The presence of MAFLD is always associated with other elements also described as related to gut microbiota changes (Figure 1). These include obesity, insulin resistance, type 2 diabetes, high blood pressure, etc. Drugs used in these situations or comorbidities can also have a profound impact on the gut microbiota, e.g., metformin, and therefore prevent the effect of pre- or probiotics targeting it [96]. It may also be possible that gut microbiota manipulation can have effects on the biotransformation of drugs taken by the patient (e.g., modification of drug metabolizing genes or enzymes in host liver or intestine) or the production of factors that may be involved in the patient’s assessment (e.g., vitamin K is also produced by the gut microbiota, and cholesterol levels are dependent on secondary bile acids that are gut microbial-derived) [97]. 

### 8.3. Is the Disease too Severe or the Trigger Still Maintained?

Changes in the microbiome are involved in the pathophysiology of MAFLD, mainly in the early stage of disease initiation (development of steatosis, early inflammatory changes, including macrophage activation) [44]. These initial alterations may therefore respond to measures targeting the gut microbiota. Later and more severe changes (such as the development of NASH, liver fibrosis or even cirrhosis) may not respond to interventions targeting the gut microbiota. However, in interventional clinical trials, patients with various stages of the disease (both early and more severe late stages) are often included and analyzed together. This is also evidenced in ALD. Some studies have shown an effect of probiotic administration in mild ethanol-induced liver injury [98,99]. However, although clear changes in the gut microbiota are also described in patients with severe alcoholic steatohepatitis, this severe disease stage does not respond to probiotics, prebiotics or antibiotics [100]. Finally, it is also conceivable that the persistence of the trigger of MAFLD (obesity and in particular visceral adiposity, unchanged dietary habits) could be stronger than concomitant attempts to modulate the gut microbiota [101]. This is why combination therapies targeting different aspects of the disease are effective in NASH [55,93]. For example, lanifibranor (a pan-peroxisome proliferator activated receptor agonist) shows very encouraging results (improvement across the whole range of liver histological lesions) by targeting not only liver metabolism and inflammation but also adipose tissue dysfunction that partly induces the liver disease [55,102]. Again, the same is true for ALD. The positive results on the use of probiotics concern patients not only with a mild disease but also following a detoxification program that allows the trigger to be stopped [98,99].

### 8.4. The Impact of an Intervention (Placebo Arm)

Multiple factors can modify the severity of the liver disease (change in diet, adaptation of physical activity or concomitant medication). These changes may be facilitated by participation in the clinical trial. A significant improvement is noted for many studies in fibrosing NASH. A placebo effect on histological improvement (reduction of NAS score by 2 points or more) is present in about 25% of patients [103]. This correlates with the number and the duration of visits, the change in body mass index and the severity of the initial disease [103]. The same placebo effect is present in studies of oral antihyperglycaemic agents [103]. This placebo effect can of course affect the results of the test intervention, especially if the effects of the test product are mild.

### 8.5. The Baseline Situation: The Responder and Non-Responder Concept

Not all patients respond in the same way to an intervention. This is also true for studies targeting the gut microbiota. Some patients respond to the intervention, sometimes very markedly, and others do not. The initial composition of the microbiota is unique to each individual and may be associated with the impact of the intervention [104]. Experiments have been carried out in this area, using the gut microbiota of responder and non-responder patients that were transplanted into antibiotic-pretreated mice and then assessing their response to the same intervention. These experiments show that the baseline gut microbiota determines the response to certain interventions, in particular the administration of inulin on weight reduction or the degree of liver steatosis [79]. The magnitude of response could therefore be influenced by a subset of bacteria (rather than one specific bacterium) simultaneously affected by prebiotics. Similarly, the use of prebiotics such as inulin is also more effective in patients with a certain physical activity threshold [105]. In the future, a personalized approach based on a precise baseline analysis could be imagined for prebiotic interventions in order to find target patients prone to having a favorable response.

## 9. Conclusions

The gut-liver axis is clearly established from extensive mechanistic studies in animals and in humans. The most promising strategies so far in humans with MAFLD are the administration of rifaximin, prebiotics and probiotics. However, the effects noted are often indirect and this is still a developing area. Targeting the gut microbiota to treat MAFLD is therefore not ready for prime time. Interestingly, it should be noted that the individual response to an intervention is variable. Certain microbial signatures seem to respond particularly well to prebiotic administration for example. Precise characterization of broad microbial changes (bacteria, viruses, fungi) in MAFLD patients according to each disease stage is therefore necessary, as well as adequate large intervention studies in targeted patients with approved and reproducible endpoints in order to unravel the relevance of microbiota interventions in the management of MAFLD.

## Figures and Tables

**Figure 1 cells-11-02718-f001:**
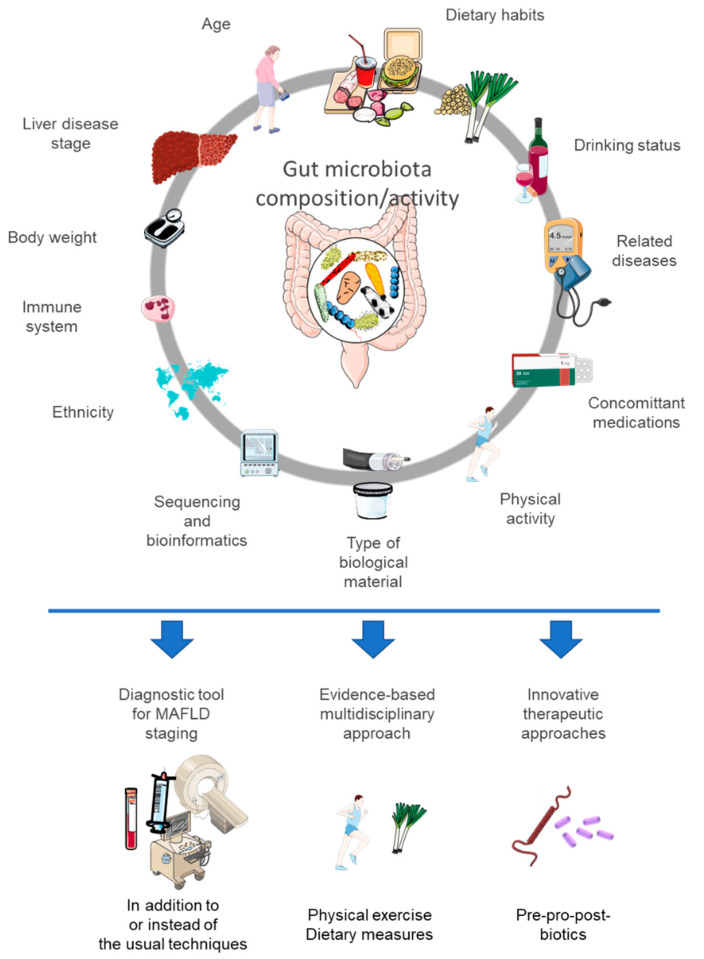
Many factors can influence the composition or activity of the gut microbiota, which can also be used or targeted in the assessment and management of metabolic dysfunction-associated fatty liver disease. This figure was partly created using Servier Medical Art templates (https://smart.servier.com) (accessed on 5 July 2022).

**Table 1 cells-11-02718-t001:** Association of microbiota and metabolic dysfunction-associated fatty liver disease (MAFLD), non-alcoholic steatohepatitis (NASH) and NASH-related fibrosis.

Disease Stage	Bacterial Microbiota Changes	Reference
MAFLD versus healthy individual controls	Changes in gut microbiota diversity	[27,28,29,30]
Gram classification	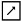 Gram negative  Gram positive	[29]
Phylum	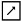 Proteobacteria  Firmicutes	[28,30,39][29,39,40]
Family	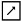 Enterobacteriaceae  Rikenellaceae, Ruminococcaceae	[30][40,41]
Genus	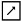 *Escherichia, Dorea, Peptoniphilus*  *Anaerosporobacter, Coprococcus, Eubacterium, Faecalibacterium, Prevotella,* *Clostridium sensu stricto*	[30,40,41][27,30,40]
Severe MAFLD or NASH versus mild MAFLD cases	Same gut microbiota diversity	[27,31,32,33]
Gram classification	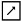 Gram negative  Gram positive	[42]
Phylum	 Firmicutes	[42]
Family	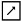 Enterobacteriaceae  Prevotellaceae  Clostridiaceae	[32][33][31]
Genus	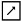 *Shigella, Bacteroides*  *Prevotella*, *Clostridium sensu stricto*	[31,33][31,33]

**Table 2 cells-11-02718-t002:** Bacterial metabolites with potential implication in metabolic dysfunction-associated fatty liver disease (MAFLD) pathogenesis.

Metabolite	Source	Involved Bacteria	Effect	Reference
Secondary bile acids LCA and DCA	Primary bile Acids	*Clostridium* sp.*Lactobacillus**Enterococcus**Bifidobacterium**Bacteroides*	More hydrophobic/toxic molecules?Easier reabsorption, reduction of gut bile salt concentration?Less FXR activation than CDCAHigher TGR5 activation than CDCA and CA	[57]
Short chain fatty acids: butyrate, propionate, acetate	Polysaccharides	*Lachnospiriaceae* fam.*Ruminococcacecae* fam.*Eubacterium rectale**Faecalibacterium prausnitzii**Roseburia* sp.*Anaerostipes*	 gut inflammation 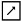 gut barrier 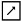 GLP-1 production 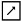  steatosis	[26]
Indole	Dietary tryptophan	*Escherichia coli*	 liver macrophage activation  gut endotoxin	[58,59,60]
Ethanol	Polysaccharides	*Klebsiella pneumoniae* *Escherichia*	Alcohol in the (portal) blood, oxidative stress and liver damage	[40,61]
Phenylacetate	Phenylalanine	Proteobacteria	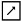 steatosis	[39]
Trimethyl-5-aminovaleric acid (TMAVA)	Trimethyllysine	*Enterococcus faecalis* *Pseudomonas aeruginosa*	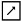 steatosis	[62]
Succinate	Polysaccharides	PrevotellaceaeVeillonellaceae	Controversial data	[63]

LCA: lithocholic acid; DCA: deoxycholic acid; CDCA: chenodeoxycholic acid; CA: cholic acid; FXR: farnesoid X receptor; TGR5: Takeda G-coupled receptor 5.

**Table 3 cells-11-02718-t003:** Clinical trials with antibiotics, probiotics, prebiotics, synbiotics or fecal microbiota transplantation in metabolic dysfunction-associated fatty liver disease (MAFLD) or non-alcoholic steatohepatitis (NASH).

Intervention	Compound and Number of Patients	Time	Disease Stage	Main Results	Reference
Antibiotic	Rifaximin (*n* = 42), before vs. after treatment	28 days	MAFLD + NASH	 ALT (in all patients)  AST (only in NASH patients)	[70]
Rifaximin (*n* = 15), before vs. after treatment	6 weeks	NASH	= ALT= HOMA-IR= steatosis (MRS)	[71]
Rifaximin (*n* = 21) vs. placebo (*n* = 21)	6 months	NASH	 ALT  AST  CK-18  liver fibrosis score  HOMA-IR	[72]
Probiotic	*Bifidobacterium*, *Lactobacillus*, *Lactococcus*, *Propionibacterium* strains (*n* = 30) vs. placebo (*n* = 28)	8 weeks	MAFLD	 fatty liver index= ALT  AST  γGT  TG	[73]
(Pasteurized) *Akkermansia muciniphila* (*n* = 12) vs. placebo (*n* = 11)	12 weeks	Overweight	 AST  γGT  insulin	[74]
Prebiotic	Oligofructose vs. placebo (*n* = 7 in total)	8 weeks	MAFLD	 AST	[75]
Oligofructose (*n* = 8) vs. placebo (*n* = 6)	36 weeks	NASH	 NAS (  steatosis)= ALT	[76]
Inulin (*n* = 9) vs. inulin-propionate ester (*n* = 9)	42 days	MAFLD + NASH?	= steatosis (vs. placebo) 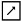 steatosis (post. vs. pre)	[77]
Inulin + inulin-rich vegetables (*n* = 75) vs. maltodextrin + inulin-poor vegetables (controls, *n* = 75)	3 months	MAFLDObesity	 AST= ALT  body weight= steatosis (controlled attenuation parameter, TE) = Fibrosis (TE)	[78]
responders	[79]
Synbiotic	*Lactobacillus*, *Bifidobacterium*, *Streptococcus thermophilus* strains + fructooligosaccharides (*n* = 38) vs. placebo (*n* = 37)	12 weeks	MAFLD	 ALT  liver stiffness  body weight	[80]
Inulin + *Bifidobacterium* (*n* = 34) vs. conventional yogurt (*n* = 34) or controls (*n* = 34)	24 weeks	MAFLD	 ALT  γGT  TG  steatosis (US)	[81]
Fructo-oligosaccharides + *Bifidobacterium* (*n* = 55) versus placebo (*n* = 49)	10–14 months	MAFLD	Steatosis = (MRS) Fibrosis = (TE)	[82]
Fecal microbiota transplantation	Allogenic FMT (*n* = 15) versus autologous FMT (*n* = 6)	One infusion	MAFLD	Steatosis = (MRI-PDFF)	[83]
FMT (*n* = 27) versus non-FMT (*n* = 20)	3-day infusion	MAFLD	Steatosis  (controlled attenuation parameter, TE)	[84]

For effect on measurable outcomes, an upward effect is denoted by (
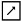
), a downward effect is denoted by (

), and no effect is denoted by (=). AST, aspartate aminotransferase; ALT: alanine aminotransferase; Chol, cholesterol; LDL-c, low-density lipoprotein cholesterol; NAFLD, nonalcoholic fatty liver disease; NAS, NAFLD activity score; NASH, nonalcoholic steatohepatitis; TG, triglycerides; US, ultrasonography; VLDL, very-low-density lipoprotein; γGT, gamma-glutamyltransferase; MRS: magnetic resonance spectroscopy; FMT, fecal microbiota transplantation; TE, transient elastography; MRI-PDFF, magnetic resonance imaging derived proton-density-fat-fraction.

## Data Availability

Not applicable.

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
