# Peer review of "Targeting the Gut Microbiome to Treat Metabolic Dysfunction-Associated Fatty Liver Disease: Ready for Prime Time?"

_cells, 2022, doi:10.3390/cells11172718_

Round 1

Reviewer 1 Report

The manuscript “Targeting the gut microbiome to treat metabolic dysfunction associated fatty liver disease: ready for prime time?” presents an arduous and rigorous path that summarizes lines of research that are of great interest to the scientific and medical community but with great complexity and multiple influencers.

I enjoyed reading it.

Please see below some points and questions I have for the authors

Lines 59-60: Microbiome definition

To my knowledge, the term refers to the entire habitat, including the microorganisms (bacteria, archaea, lower and higher eukaryotes, and viruses), their genomes (i.e., genes), and the surrounding environmental conditions.

Marchesi, J.R., Ravel, J. The vocabulary of microbiome research: a proposal. Microbiome 3, 31 (2015). https://doi.org/10.1186/s40168-015-0094-5

I am not sure what the authors define as the microbiome.

Table 1 is a bit confusing in this way.

References are also missing

Subsection:8.3. Is the disease too severe or the trigger still maintained?

Sorry, but after reading twice this paragraph, it is not fully understood by this reviewer

Author Response

The manuscript “Targeting the gut microbiome to treat metabolic dysfunction associated fatty liver disease: ready for prime time?” presents an arduous and rigorous path that summarizes lines of research that are of great interest to the scientific and medical community but with great complexity and multiple influencers. I enjoyed reading it. Please see below some points and questions I have for the authors

Lines 59-60: Microbiome definition

To my knowledge, the term refers to the entire habitat, including the microorganisms (bacteria, archaea, lower and higher eukaryotes, and viruses), their genomes (i.e., genes), and the surrounding environmental conditions.

Marchesi, J.R., Ravel, J. The vocabulary of microbiome research: a proposal. Microbiome 3, 31 (2015). https://doi.org/10.1186/s40168-015-0094-5

I am not sure what the authors define as the microbiome.

Answer:

We have been clearer on the definition of the microbiome. In order to standardize the emergent studies focusing on the microbiome, a panel of experts recently revisited the microbiome definition (Berg et al. 2020). In this new definition, the microbiome encompasses both the microorganisms, including viruses, bacteria, archaea, unicellular eukaryotes, and fungi, and their “theatre of activity” (structural elements, metabolites/signal molecules, and the surrounding environmental conditions).

The text has therefore been amended to include this reference.

We decided to refer to the microbiome rather than the microbiota in the title because of the data we present on the metabolites produced by microbes as well as the data on the use of postbiotics.

We have also added some information about fungi, viruses and archaea.

Table 1 is a bit confusing in this way. References are also missing.

Answer:

A lot of information is indeed present in this table. We have therefore decided to lighten it by removing the information on suspected mechanisms (detailed later in the text) and to replace the column with a column of references, as you suggest.

Subsection:8.3. Is the disease too severe or the trigger still maintained?

Sorry, but after reading twice this paragraph, it is not fully understood by this reviewer.

Answer:

Thank you for your comment. We have rewritten the content and hope that this paragraph is clearer.

Reviewer 2 Report

The Authors should analyze te role of endotoxin in the pathogenesis of MAFLD. In addition they should discuss the importance of oxidant stress, previously related to endotoxin in NAFLD. They have to add a new chapter, analyzing markers of oxidant stress, such as 8-iso-PGF2alpha and sp-NOX2, overexpressed in NAFLD.

Author Response

The Authors should analyze the role of endotoxin in the pathogenesis of MAFLD. In addition they should discuss the importance of oxidant stress, previously related to endotoxin in NAFLD. They have to add a new chapter, analyzing markers of oxidant stress, such as 8-iso-PGF2alpha and sp-NOX2, overexpressed in NAFLD.

Answer:

Thank you for your comment. We have added the name "endotoxin" in the text and added a paragraph on oxidative stress and the markers you mention. This paragraph is not separated from the one on LPS and PAMPs because it is the same entity.

Round 2

Reviewer 2 Report

The Authors answered correctly to my queries. The paper has been sensibly improved.